# PhysMaster: Mastering Physical Representation for Video Generation via Reinforcement Learning

## Abstract

Video generation models nowadays are capable of generating visually realistic videos, but often fail to adhere to physical laws, limiting their ability to generate physically plausible videos and serve as "world models". To address this issue, we propose PhysMaster, which captures physical knowledge as a representation for guiding video generation models to enhance their physics-awareness. Specifically, PhysMaster is based on the image-to-video task where the model is expected to predict physically plausible dynamics from the input image. Since the input image provides physical priors like relative positions and potential interactions of objects in the scenario, we devise PhysEncoder to encode physical information from it as an extra condition to inject physical knowledge into the video generation process. The absence of physical representation definition prevents a straightforward supervision of PhysEncoder. Thus we leverage the physical plausibility of generated videos guided by PhysEncoder as feedback for reinforcement learning (RL), thereby optimizing PhysEncoder for extracting physical representations with Direct Preference Optimization (DPO). PhysMaster provides a feasible solution for improving physics-awareness of PhysEncoder and thus of video generation, proving its ability on a simple proxy task and generalizability to wide-ranging physical scenarios. This implies that our PhysMaster, which unifies diverse physical processes via RL-based representation learning, offers a generic and plug-in solution for physics-aware video generation and broader applications.

## 1 Introduction

Video generation models (Brooks et al., 2024; Kuaishou, 2024; Yang et al., 2024c) have developed rapidly nowadays, achieving significant performances in generating visually appealing videos (RunwayML, 2024; Team, 2024; Kong et al., 2024). However, they primarily act as sophisticated pixel predictors based on case-specific imitation, and often face challenges in adherence to physical laws (Kang et al., 2024; Liu et al., 2025a; Meng et al., 2025). This limits their ability to generate physically plausible videos and further comprehend physical principles to serve as "world models". To evolve these models from content creators to world simulators, we aim to incorporate physical knowledge into the video generation process to enhance their physical realism.

Solutions for physics-aware video generation can be broadly categorized into two types based on the usage of simulation. Simulation-based approaches (Lv et al., 2024; Liu et al., 2024b) attempt to apply physics-based simulation results to guide video generation, but they are often constrained in the range of simulable physical processes and modalities, lacking the potential to generalize to diverse phenomena. Simulation-free methods (Xue et al., 2024; Furuta et al., 2024) rely on post-training on physics-rich data or employ reinforcement learning for aligning to human preference. The former highly depends on fitting similar training samples, and the latter utilizes either expensive human annotation suffering from rater variability, or scalable but inaccurate AI evaluators. In summary, existing works find it hard to truly abstract and understand physics of the world (Lin et al., 2025; Motamed et al., 2025), hindering generalization to diverse physics. We summarize the specific challenges of physics-aware video generation into the following two points. First, the commonly used Mean Squared Error (MSE) loss for data-driven finetuning focuses on appearance fitting rather than comprehension of physical knowledge, and it is non-trivial to directly supervise the physical per-

formance of pretrained models beyond merely appearance. Second, generative models struggle to extract appropriate physical knowledge from a textual instruction or an input image and translate it into physical guidance for generation, which demands logical reasoning from descriptions or images to physical knowledge, and to visual phenomena.

Facing the aforementioned challenges, we propose to learn a physical representation as the bridge between the necessary physical knowledge and generated videos to guide generative models towards physics awareness. Specifically, we focus on image-to-video (I2V) generation where an initial frame and textual description are given, the model is supposed to predict physically plausible dynamics from input scenes. The input image offers visual cues like object configurations, relative positions, and potential interactions that largely dictate the subsequent physical evolution of video, making it a reliable source of physical priors. Thus firstly, we devise a physical encoder, PhysEncoder, to extract implicit physical representation from the input image as an extra input condition to guide the generation process for enhancing physics-awareness of the model.

However, how to learn an appropriate physical representation for video generation remains an open question. Without a explicit definition for physical representation, we can not conduct straightforward supervision of PhysEncoder. Thus we propose indirect supervision by optimizing its ability to guide physically plausible video generation. This is achieved through a reinforcement learning with human feedback (RLHF) framework, which has proven effective in finetuning of both large language models (LLMs) (Yuan et al., 2023; Xu et al., 2024; Yuan et al., 2023) and generative models (Lee et al., 2023; Prabhudesai et al., 2024; Fan et al., 2023). The physical plausibility of generated videos guided by PhysEncoder acts as feedback for optimizing PhysEncoder to extract effective physical representations. Specifically, we train PhysEncoder on human preference data via Direct Preference Optimization (DPO) (Rafailov et al., 2023) in a three-stage training pipeline. We first conduct supervised fine-tuning (SFT) of both base model and PhysEncoder, then we adopt a two-stage DPO with pairwise supervision to enhance PhysEncoder's capacity to capture physical representations and model's physical performance, with trainable module separately set as LoRA (Hu et al., 2021) of the DiT model and PhysEncoder. With SFT providing the model with the initial ability to predict physically plausible videos under the guidance of simultaneously finetuned PhysEncoder, the subsequent DPO processes further steer PhysEncoder's output towards physics-aware representation, thus helping improve the physical understanding of model.

Last but not least, we condition the video generation model on physical representation in a plug-in manner, enhancing physics-awareness by injecting physical knowledge into it. Such a paradigm enables the model to learn general physical properties, rather than overfitting to specific phenomena or being constrained to particular motion modalities as in previous works, thus allowing it to generalize to diverse scenarios. We demonstrate the effectiveness of PhysEncoder starting from a simple proxy task of "free-fall", and then generalize to broader physical scenarios governed by a wide range of physical laws. PhysEncoder proves its capablity by guiding the generation model towards enhanced physical performance, and such generalization implies that PhysMaster, our representation learning paradigm facilitates the physical understanding of physical laws in a broad scope.

In summary, PhysMaster provides a more generalizable solution for video generation models to capture physical knowledge across diverse physical phenomena, showing its advantage in acting as a foundational solution for physics-aware video generation and potential to energize more fancy applications (Agarwal et al., 2025; Yang et al., 2024b; 2023).

## 2 RELATED WORKS

**Physics-aware video generation.** While recent video generation models achieve impressive visual effects (Brooks et al., 2024; Kong et al., 2024), they still struggle with adherence to real-world physical laws (Lin et al., 2025; Kang et al., 2024). Physics-aware video generation approaches can be broadly categorized based on the application of explicit physical simulation. Simulation-based methods (Lv et al., 2024; Xie et al., 2025; Montanaro et al., 2024; Zhang et al., 2024b) guide generation with simulation results. PhysGen (Liu et al., 2024b) utilizes rigid-body dynamics simulated with physical parameters inferred by large foundation models. PhysMotion (Tan et al., 2024) relies on MPM-based simulation to generate coarse videos which are refined by a video diffusion model. As for simulation-free approaches, they either fine-tune on large-scale video datasets to implicitly internalize physical priors (Wang et al., 2025; Zhang et al., 2025), or use reinforcement learning with

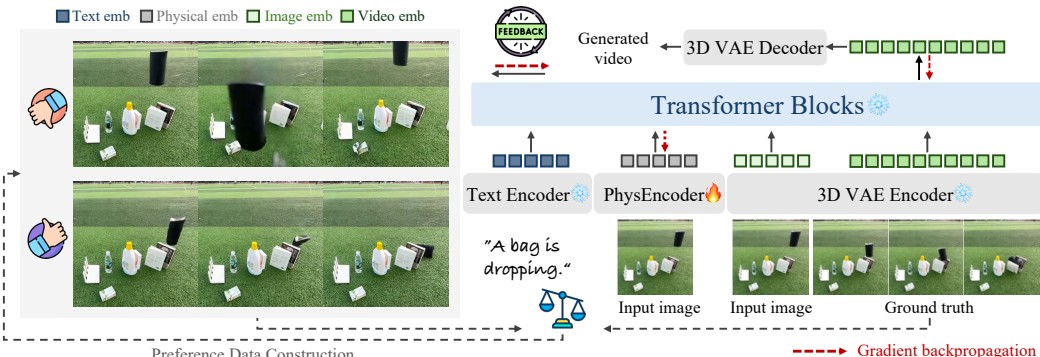

Figure 1: **Overall architecture of PhysMaster.** Given an input image, PhysEncoder encodes its physical feature and concatenates with visual features, then the DiT model predicts subsequent frames conditioned on physical, visual, and text embeddings. We optimize PhysEncoder 's physical representation via feedback from generated video pairs of the model by maximizing reward derived from "positive" and "negative" video outputs in a DPO paradigm.

feedback from human annotators or vision-language models (Xue et al., 2024; Furuta et al., 2024). PhyT2V (Xue et al., 2024) uses MLLMs to refine prompts iteratively through multiple rounds of generation and reasoning. WISA (Wang et al., 2025) incorporates structured physical information into the generative model and uses Mixture-of-Experts for different physics categories. However, those methods are restricted to fixed physical categories or exhibit limited physical comprehension. Our PhysMaster incorporates physical knowledge into video generation process via physical representation to enhance general physics-awareness.

**RLHF for video generation.** Inspired by the success of RLHF in LLMs (Ouyang et al., 2022; Jaech et al., 2024), researchers have explored applying this paradigm to video generation (Zhang et al., 2024a; Qian et al., 2025). VideoDPO (Liu et al., 2024a) pioneers the adaptation of DPO (Rafailov et al., 2023) to video diffusion models by considering both visual quality and semantic alignment for data pair construction. VideoAlign (Liu et al., 2025b) introduces a multi-dimensional video reward model and DPO for flow-based video generation model based on it. PISA (Li et al., 2025) investigates specifically for video generation of object free-fall, improving physical accuracy through reward modeling based on depth and optical flow. Unlike the aforementioned methods, we optimize a physical encoder rather than the whole video generation model by leveraging generative feedback from the model's outputs. This paradigm mitigates overfitting to specific physical processes and promotes the encoder's generalizability for learning universal physical knowledge through RLHF.

## 3 METHOD

Based on I2V setting, PhysMaster extracts physical representation from the input image and optimizes both the generation model and PhysEncoder in a three-stage training pipeline. It seeks direct supervision from groundtruth via SFT and pairwise supervision from generated videos via DPO, and is implemented on the simplest proxy task and broader scenarios. We will separately detail physical representation (Sec 3.1), task formulation (Sec 3.2) and training scheme (Sec 3.3).

### 3.1 PHYSICAL REPRESENTATION

PhysMaster is implemented upon a transformer-based diffusion model (DiT) (Peebles & Xie, 2023), which employs 3D Variational Autoencoder (VAE) (Kingma, 2013) to transform videos and initial frame to latent space, and T5 encoder $\mathcal{E}_{T5}$ (Raffel et al., 2020) for text embeddings $c_{text}$.

We propose to learn a physical representation from input image as extra guidance for the I2V model to inject physical information, since the input image contains not only explicit physical states, such as object material and spatial distribution, but also implicit physical laws, like the gravitational field. It is worth expecting that the learned physical representation can be used as a generalizable guidance of both physical properties and dynamics for physics-aware video generation. Following the structure of Depth Anything (Yang et al., 2024a), we build PhysEncoder with a DINOv2 (Oquab et al.,

2023) encoder and a physical head. The former adopts pretrained weights from Yang et al. (2024a) for initialization and takes the role of semantic perception, while the latter adapts the extracted high-level semantic features into an appropriate dimension to be injected into the DiT model. Taking the first frame as image input, PhysEncoder encodes it into physical embeddings $c_{phys}$, which are then fed into DiT model after concatenated with image embeddings $c_{image}$. For SFT, the flow-based DiT model with weights $\theta$ directly parameterizes the $v_\theta(z_t, t, c_{text}, c_{image}, c_{phys})$ to regress velocity $(z_1 - z_0)$ with the Flow Matching objective (Lipman et al., 2022):

$$\mathcal{L}_{LCM} = \mathbb{E}_{t, z_0, \epsilon} ||v_\theta(z_t, t, c_{text}, c_{image}, c_{phys}) - (z_1 - z_0)||_2^2. \tag{1}$$

## 3.2 TASK FORMULATION

Our work aims to provide a scalable and generalizable methodology for learning physics from targeted data, so for demonstrating the effectiveness of our PhysMaster, we start by defining a proxy task under simple physical principles and construct domain-specific data for preliminary validation; then we verify its generalizability across a broader range of physical laws and various tasks.

**Proxy task.** For preliminary verification, "free-fall" (involving the complete physical process of objects dropping from mid-air and colliding with other objects on a surface), a simple yet expressive scenario is chosen as the proxy task for the following characteristics. First, "free-fall" embodies clear and fundamental physical principles (e.g., energy and momentum conservation) shared across diverse physical scenarios, making it a suitable representative for further generalization. Second, such physical scenario involves a wide range of object-level physical properties, such as density, elasticity, and hardness, allowing proof of generalizability of learned representations across different physical attributes. Third, this task can be easily simulated for scalable generation of synthetic data and allows for straightforward evaluation by comparing generated videos against ground-truths. The reason is that by assuming the falling object starts from rest and is only influenced by gravity, the trajectories of objects become fully deterministic given the initial frame, which also enables automatic construction of preference video pairs for DPO by similarity with ground-truths.

**Broader scenarios.** We further substantiate generalization capabilities of PhysMaster across diverse physical processes. Following WISA (Wang et al., 2025), we include large-scale scenarios broadly covering common physical phenomena observed in real world for PhysEncoder to acquire a far more comprehensive and generalizable understanding of physical laws and thus effectively enhances the physics awareness of the video generation model. Different from the proxy task implementation, we modify the text prompts provided to the generation model by adding domain-specific prefixes (e.g., "*Optic*, A ray of light ...", "*Thermodynamic*, A glass of water ..."). This conditions the model on the type of involved physics laws and guides it to associate visual phenomena with underlying physics extracted from PhysEncoder. For preference assignment, we rely on human annotators to provide pairwise labels for DPO data construction and evaluation.

## 3.3 TRAINING SCHEME

We propose a three-stage training pipeline for PhysMaster to enable physical representation learning of PhysEncoder by leveraging the generative feedback from I2V model. The core idea is formulating DPO for PhysEncoder with the reward signal from generated videos of pretrained DiT model, thus help physical knowledge learning without explicit modeling.

**Stage I: SFT for DiT and PhysEncoder.** First, we condition the I2V base model on physical representation from PhysEncoder by SFT, thus it is possible for us to optimize PhysEncoder with the performance of model as feedback in following stages. Since PhysEncoder's training starts from the frozen DINOv2 with pretrained weights from Depth Anything (Yang et al., 2024a) and trainable physical head with randomly initialized weights, this stage can be viewed as adapting Depth Anything for physical condition injection. As in Figure 1, by concatenating physical embeddings extracted by PhysEncoder with visual embeddings encoded by VAE, we inject physical representation as extra condition to the model. SFT following Eq 5 equips the model with the initial ability to predict subsequent frames from the input image, guided by simultaneously finetuned PhysEncoder.

**Stage II: DPO for DiT.** Second, we expect to adapt the output of the pretrained model to a more physically plausible distribution, paving the way for the PhysEncoder to learn from generated videos with higher physical accuracy. Then in Stage II, we apply LoRA (Hu et al., 2021) to finetune the DiT

model on preference dataset with DPO, during which the model learns to generate positive samples with higher probability and negative samples with lower probability. Regarding I2V setting, each sample in our preference dataset includes a prompt $p$, an image $i$, a human-chosen video $x^w$ and a human-rejected video $x^l$. The goal of DPO is to learn a conditional distribution $\pi_\theta(x \mid p, i)$ that maximizes the reward $r_\phi(x, p, i)$ while staying close to a reference model $\pi_{\text{ref}}$:

$$\max_{\pi_\theta} \mathbb{E}_{p \sim \mathcal{D}_p, i \sim \mathcal{D}_i, x \sim \pi_\theta(x|p,i)} \left[ r_\phi(x, p, i) \right] - \beta \, \mathbb{D}_{\text{KL}} \left[ \pi_\theta(x \mid p, i) \, \| \, \pi_{\text{ref}}(x \mid p, i) \right] \tag{2}$$

where $\beta$ controls the regularization term (KL-divergence) from $\pi_{\text{ref}}$. For our flow-based DiT model, Flow-DPO objective (Liu et al., 2025b) $L_{\text{FD}}(\theta)$ is then given by:

$$-\mathbb{E} \left[ \log \sigma \left( -\frac{\beta}{2} \left( \|v^w - v_\theta(x_t^w, t)\|^2 - \|v^w - v_{\text{ref}}(x_t^w, t)\|^2 - \left( \|v^l - v_\theta(x_t^l, t)\|^2 - \|v^l - v_{\text{ref}}(x_t^l, t)\|^2 \right) \right) \right) \right],$$
$$\tag{3}$$

where the conditioning prompt $p$ and image $i$ are omitted for simplicity, $v_\theta$ denotes the predicted velocity field, $v^w$, $v^l$ are the target velocity of "preferred" and "less preferred" data.

The pretrained DiT model from Stage I is regarded as the reference model and is used to construct the preference data pairs. Specifically, we generate two groups of videos with the pretrained model using the same prompt $p$ and initial frame $i$ but different seeds. By establishing clear distinctions between positive samples $x^w$ and negative samples $x^l$, the model learns to generate physically plausible videos. As a result, we further enhance the model's physics awareness.

**Stage III: DPO for PhysEncoder.** We leverage generative feedback from the pretrained DiT model to optimize PhysEncoder's physical representation via DPO paradigm. As illustrated in Figure 1, our framework consists of two parts: PhysEncoder to be optimized and the pretrained DiT model providing generative feedback. With physical head of PhysEncoder the only trainable module, Stage III shares the same training objective Eq 3 with Stage II, differing solely in the learnable parameters. $L_{\text{FD}}(\theta)$ leads PhysEncoder to learn a physical representation that guides the predicted velocity field $v_\theta$ closer to the target velocity $v^w$ of the "preferred" data. In this manner, by directing the DiT model to generate more accurate physical dynamics, the PhysEncoder's original representation is gradually optimized with more physical knowledge through model feedback.

## 4 EXPERIMENTS

To evaluate the effectiveness of PhysMaster for physical representation learning and demonstrate its potential to enhance physical performance of the DiT model, comprehensive experiments are conducted on both the proxy task and wide-ranging scenarios.

### 4.1 IMPLENTATION DETAILS

**Training configuration.** The training of both PhysEncoder and the DiT model is conducted on 8 NVIDIA-A800 GPUs in all three stages, with 20 hours for SFT, 15 hours for DPO on LoRA and 8 hours for DPO on PhysEncoder. The training process employs the Adam optimizer (Kingma, 2014), and we utilize 50 DDIM steps (Song et al., 2020) and set the CFG scale to 7.5 during inference.

**Dataset construction.** For the proxy task, we follow PISA (Li et al., 2025) to use Kubric (Greff et al., 2022) to create synthetic datasets of "free-fall". The object assets are sourced from the Google Scanned Objects (GSO) dataset (Downs et al., 2022). For generalizability demonstration, we utilize WISA-80K (Wang et al., 2025) encompassing 17 types of real-world physical events across three major branches of physics (Dynamics, Thermodynamics, and Optics).

**Evaluation protocols.** PisaBench (Li et al., 2025) is introduced to evaluate our model's performance on the proxy task. We use SAM 2 (Ravi et al., 2025) for segmentation of object masks and compute the following metrics between corresponding masks of generated and ground truth videos for evaluation: L2 distance between the centroids of the masked regions, chamfer distance (CD) and

Table 1: **Ablation study for models from different training stages and strategies** on proxy task, evaluated on the test set split into "seen" and "unseen". $v_\theta$ is DiT model, $E_p$ is PhysEncoder.

| Training Stages | Seen | | | Unseen | | | Average | | |
|---|---|---|---|---|---|---|---|---|---|
| | L2($\downarrow$) | CD($\downarrow$) | IoU($\uparrow$) | L2($\downarrow$) | CD($\downarrow$) | IoU($\uparrow$) | L2($\downarrow$) | CD($\downarrow$) | IoU($\uparrow$) |
| Base | 0.1066 | 0.323 | 0.119 | 0.1065 | 0.339 | 0.111 | 0.1066 | 0.331 | 0.115 |
| SFT for $v_\theta$ | 0.0532 | 0.134 | 0.137 | 0.0512 | 0.133 | 0.135 | 0.0522 | 0.134 | 0.136 |
| SFT for $v_\theta$ & $E_p$ (Stage I) | 0.0568 | 0.141 | 0.137 | 0.0498 | 0.128 | 0.134 | 0.0533 | 0.134 | 0.135 |
| SFT for $v_\theta$ + DPO for $v_\theta$ | 0.0560 | 0.143 | **0.144** | **0.0446** | **0.115** | 0.128 | 0.0503 | 0.129 | 0.136 |
| SFT for $v_\theta$ & $E_p$ + DPO for $v_\theta$ (Stage II) | 0.0520 | 0.125 | 0.133 | 0.0454 | 0.116 | 0.136 | 0.0487 | 0.120 | 0.134 |
| SFT for $v_\theta$ & $E_p$ + DPO for $E_p$ | 0.0559 | 0.140 | 0.138 | 0.0503 | 0.129 | 0.134 | 0.0531 | 0.134 | 0.136 |
| SFT for $v_\theta$ & $E_p$ + DPO for $v_\theta$ & $E_p$ | 0.0501 | 0.124 | 0.141 | 0.0458 | 0.121 | 0.140 | 0.0480 | 0.123 | 0.141 |
| SFT for $v_\theta$ & $E_p$ + DPO for $v_\theta$ + DPO for $E_p$ (Stage III) | 0.0489 | **0.120** | 0.141 | 0.0450 | **0.115** | **0.145** | 0.0470 | **0.118** | **0.143** |
| SFT for $v_\theta$ & $E_p$ + DPO for $v_\theta$ + DPO for $v_\theta$ & $E_p$ | **0.0477** | 0.121 | **0.144** | 0.0452 | **0.115** | 0.141 | **0.0466** | **0.118** | 0.142 |

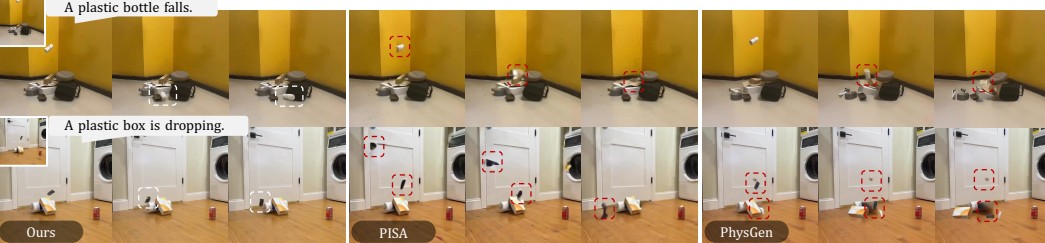

Figure 2: **Qualitative comparison with video generation models specialized for rigid-body motion** proves the advantage of our model in shape consistency and trajectory accuracy on "free-fall".

Intersection over Union (IoU) of the mask regions. We utilize VIDEOPHY (Bansal et al., 2024) for evaluating physics awareness of video generation in broader scenarios. We test on 344 carefully crafted prompts from it, which reflect a wide array of physical principles, and report the physical commonsense (PC) and semantic adherence (SA) scores.

### 4.2 EVALUATION ON PROXY TASK

To validate that our training pipeline can effectively improve the physical performance of base model on the proxy task, we compare the physical accuracy of our model on "free-fall" motion with existing works and ablate different training techniques of PhysEncoder.

**Comparison.** We compare our model with PhysGen (Liu et al., 2024b) and PISA (Li et al., 2025) on the real-world subset from PisaBench (Li et al., 2025) which is unseen to any model during training for robust evaluation. We apply more rigorous metrics of similarity over all objects in the scene by comparing against the ground truth. Table 2 shows that our model outperforms both baselines. PhysGen struggles with accurately modeling spatial relationships between objects and surfaces like ground or tables, thus often leads to physically implausible object interactions. For PISA, its best variant with depth-based reward optimizes for trajectory accuracy (comparable L2/CD) at the cost of shape consistency (lower IoU). In contrast, ours excels in IoU while maintaining competitive trajectory accuracy, achieving the best overall performance, which is also proved in Figure 2.

**Ablation study.** We report the qualitative results from different training stages and pipelines on the synthetic subset of PisaBench in Table 1, where block 1 denotes the I2V base model; block 2 - 4 refer to our model and its variants in Stage I - III of training pipeline. "Seen" corresponds to a split of videos with objects and backgrounds seen during training, and "Unseen" with novel objects and backgrounds. *1) Ablation for different training stages.* row 3, 5, and 8 indicate that our SFT endows the model with preliminary ability to predict objects' motion of "free-fall", the subsequent DPO for DiT model further steers the generated videos' distribution towards physically plausible paths, and the optimization of PhysEncoder in the last stage improves its capability in guiding model towards higher level of physics-awareness. The qualitative results in Figure 3 consistently prove our pipeline's efficacy. *2) Ablation for PhysEncoder.* The comparative pipeline in row 2 and 4 is not equipped with PhysEncoder, thus SFT and DPO are both implemented on the DiT model. Although the performance after SFT on both DiT and PhysEncoder (row 3) is even worse than SFT on DiT alone (row 2), showing that simple SFT cannot help PhysEncoder learn appropriate physical representations for guiding the DiT model towards physics-awareness, DPO unlocks PhysEncoder 's potential to extract physical information and guide the model to generate videos with better physical

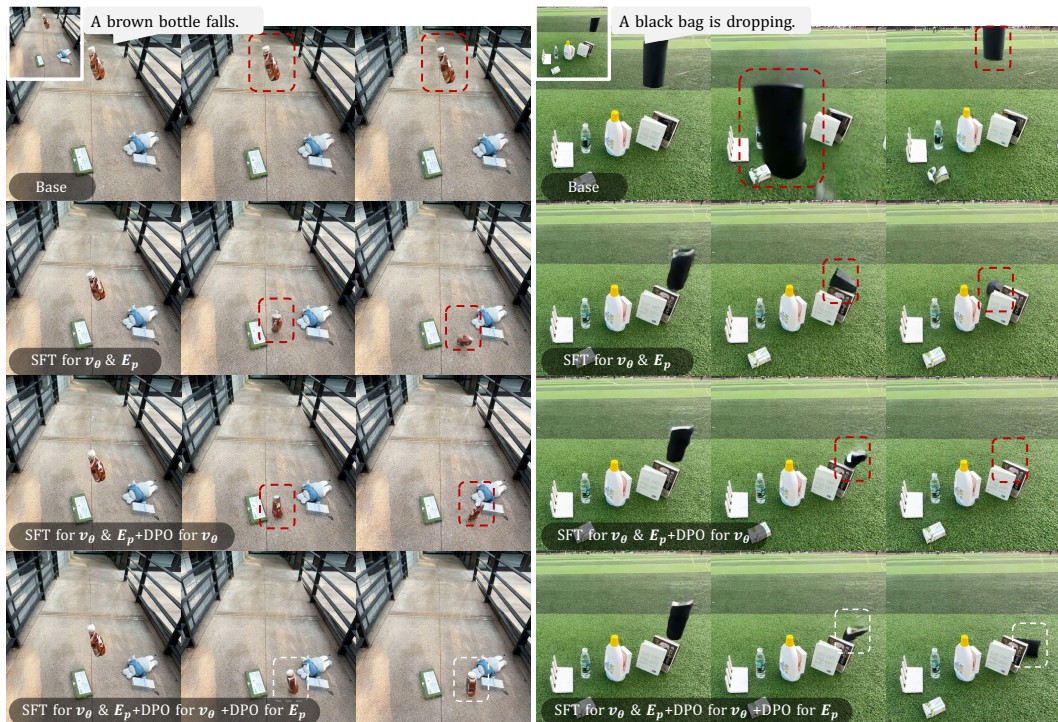

Figure 3: **Qualitative ablation for models in each training stage** on proxy task. The model exhibits a preliminary capability for predicting object motion trends after SFT. Two-stage DPO further improves model performance in preserving objects' rigidity and complying with physical laws (e.g., gravitational acceleration and collision). $v_\theta$ is DiT model and $E_p$ is PhysEncoder.

Table 2: **Quantitative comparison with video generation models specialized for rigid-body motion** verifies superiority of our model on proxy task.

| Methods | L2(↓) | CD(↓) | IoU(↑) |
|---------|-------|-------|--------|
| PhysGen | 0.0433 | 0.0967 | 0.418 |
| PISA | **0.0294** | 0.0570 | 0.433 |
| Ours | 0.0299 | **0.0567** | **0.468** |

Figure 4: **Visualization of the first three PCA components of physical representation.** $E_p$ in Stage III reveals similarities in objects under the same external forces (red: on the ground; green: in the air) over $E_p$ **in Stage I**.

performance (row 4, 5,8). **3) Ablation for DPO strategies.** All strategies of DPO succeed in further improving physical accuracy on average than previous stages. Only optimizing PhysEncoder (row 6) encounters difficulties in performance improvements. The model itself has not been aligned to adherence to physics before providing feedback to PhysEncoder, preventing DPO from functioning effectively. Although Stage II of our training pipeline underperforms joint DPO of DiT and PhysEncoder (row 7), our Stage III surpasses all other methods. Joint optimization of DiT and PhysEncoder in Stage III (row 9) achieves comparable overall performances with our Stage III but performs worse on "unseen" split, probably because this variant with trainable DiT is more likely to overfit training data, harming the model's generalizability to novel scenarios.

**PCA analysis.** We also visualize the principal component analysis (PCA) on the physical features from PhysEncoder in Stage I and Stage III in Figure 4. In our Stage III physical feature maps, similarities are shown clearly for objects under the same external forces, (green for objects in mid-air and only affected by gravity, red for objects on the ground subjected to the support force); differences are also shown more obviously between materials (e.g., deformable object in white box has clearly distinct colors), which proves two aspects of physical understanding of our PhysEncoder.

## 4.3 GENERALIZATION ON BROADER SCENARIOS

PhysEncoder demonstrates its physics-awareness for enhancing model's physical realism on the proxy task, suggesting its potential to generalize to broader scenarios. We apply our training pipeline

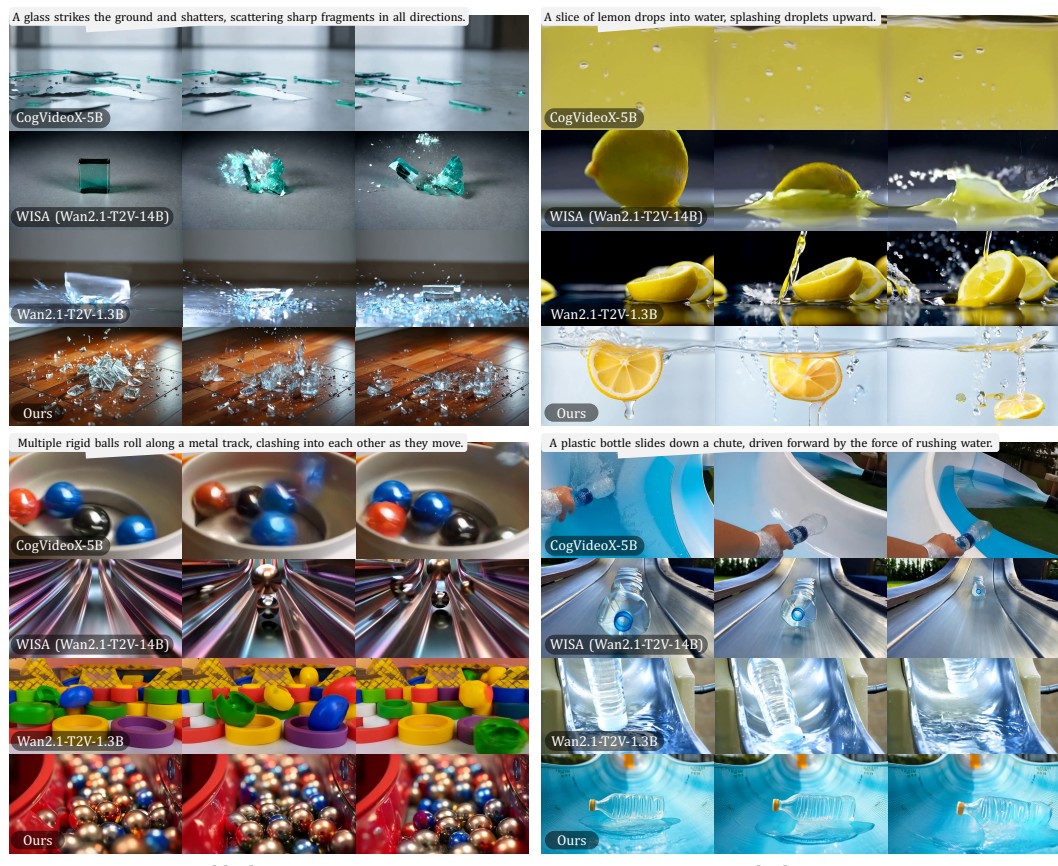

Figure 5: **Qualitative comparison with existing T2V models on broader scenarios** including objects of various materials and in different environments, validates the generalizability of our method.

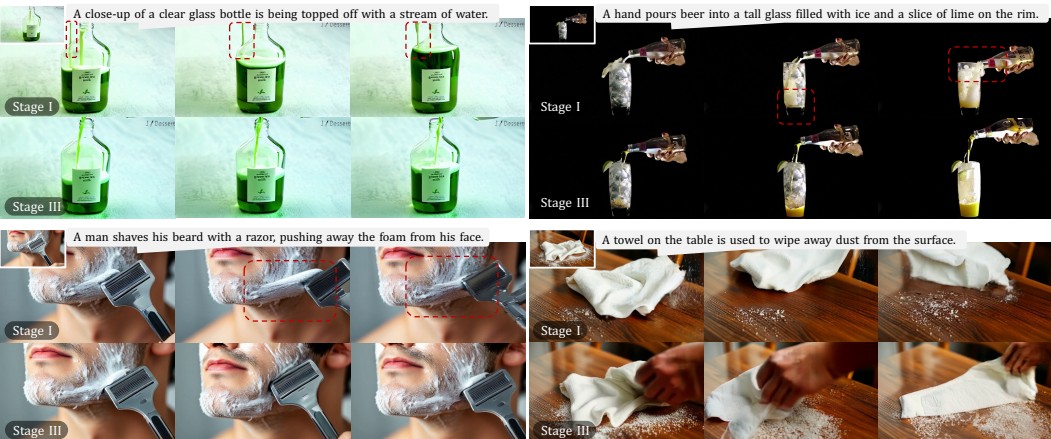

Figure 6: **Qualitative ablation for models in different stages** on broader scenarios. DPO following Stage I improves the physical coherence of model in Stage III (e.g., fluid mechanics and gravitation).

on a large-scale dataset (Wang et al., 2025) broadly covering common physical phenomena observed in real world to substantiate the generalizability of our method.

**Comparison.** We compare with two types of video generation models, general models including HunyuanVideo (Kong et al., 2024), CogVideoX-5B (Yang et al., 2024c), Cosmos-Diffusion-7B (Agarwal et al., 2025), Wan2.1-T2V-1.3B and specialized physics-focused models represented by PhyT2V (Xue et al., 2024) and WISA (Wang et al., 2025). Table 3 shows that, although our

Table 3: **Quantitative comparison with existing video generation models** on broader scenarios . Our model shows superior performance in both physics-awareness and efficiency.

| Methods | Inference Time(s) | SA(↑) | PC(↑) |
|---|---|---|---|
| HunyuanVideo | 1080 | 0.46 | 0.28 |
| Wan2.1-T2V-1.3B | 180 | 0.49 | 0.24 |
| CogvideoX-5B | 210 | 0.60 | 0.33 |
| Cosmos | 600 | 0.57 | 0.18 |
| PhyT2V | 1800 | 0.61 | 0.37 |
| WISA | 220 | **0.67** | 0.38 |
| Our base model | **23** | 0.59 | 0.29 |
| Our final model | 26 | **0.67** | **0.40** |

Table 4: **Ablation study for PhysEncoder** on broader scenarios. $v_\theta$ is DiT model, $E_p$ is PhysEncoder. The results validate that DPO enables $E_p$ to acquire a comprehensive understanding of real-world physics and thus effectively enhances the physics awareness of $v_\theta$.

| Methods | SA(↑) | PC(↑) |
|---|---|---|
| Base | 0.59 | 0.29 |
| SFT for $v_\theta$ | 0.63 | 0.33 |
| SFT for $v_\theta$ + DPO for $v_\theta$ | 0.64 | 0.35 |
| SFT for $v_\theta$ & $E_p$ (Stage I) | 0.61 | 0.33 |
| SFT for $v_\theta$ & $E_p$ + DPO for $v_\theta$ + DPO for $E_p$ (Stage III) | **0.67** | **0.40** |

base model is surpassed by CogvideoX-5B, the base model of WISA, our final model in Stage III achieves state-of-the-art performance on both SA and PC metrics, demonstrating that our proposed method enhances the realism of generated videos physically and semantically. Our model also has a significant advantage in efficiency. It is approximately 70x faster than PhyT2V—an iterative method requiring feedback from VLM, and 8x faster than WISA. Our model generates a 5-second video in just 26 seconds on a single A800 GPU, establishing it as a highly practical solution that does not sacrifice physical or semantic adherence. Figure 5 includes qualitative comparison with both existing T2V models, demonstrating our superior ability in challenging cases of both rigid-body and fluid motion.

**Ablation study.** We conduct ablation analysis to verify the effectiveness of our core component and strategy of training in Table 4. *1) Effectiveness of PhysEncoder:* Compared to our base model (row 1), our final model (row 5) improves SA and PC scores by 0.08 and 0.11. The comparative pipeline is not equipped with PhysEncoder, with SFT (row 2) and the following DPO (row 3) both implemented on the DiT model only. Such a pipeline without PhysEncoder improves SA and PC scores by 0.05 and 0.06, proving the advantage of our proposed PhysEncoder in successfully extracting crucial physical knowledge from the training data and using it to guide the generator toward greater physical realism, which is unattain-

Table 5: **User study for models from different stages** validates the effect of our training pipeline in two selected physical scenarios. Our final model shows superior ability of PhysEncoder in Stage III in enhancing the model's physics-awareness over the base model and Stage I.

| Methods | Rigid-body movement | Fluid motion |
|---|---|---|
| Base | 7.8 | 12.2 |
| Stage I | 25.3 | 16.7 |
| Stage III | **66.9** | **71.1** |

able by simply applying SFT or DPO to the DiT model alone. *2) Effectiveness of DPO:* Simply applying SFT to PhysEncoder (row 2 vs. row 4) does not yield an immediate benefit, suggesting that SFT alone is insufficient for PhysEncoder to learn a useful guiding physical representation. However, DPO unlocks the potential of PhysEncoder, allowing it to effectively translate its learned physical representations into improved generation quality in both physical commonsense and semantic adherence (row 4 vs. row 5). Additionally, Figure 6 visualizes the videos generated by our model in Stage I and III, further validating the effectiveness of DPO. *3) Effectiveness of whole training pipeline:* Table 5 includes human preference rates among models from different training stages in two types of real-world scenarios, showing that annotators prefer videos from Stage III model than Stage I model or I2V base model in adherence to physical laws.

# 5 CONCLUSIONS

We propose PhysMaster, which learns physical representation from input image for guiding I2V model to generate physically plausible videos. We optimize physical encoder based on generative feedback from a pretrained video generation model via DPO, which proves to enhance the model's physical accuracy and demonstrate generalizability across various physical processes by injecting physical knowledge into generation, proving its potential to act as a generic and plug-in solution for physics-aware video generation and broader applications.

**Limitations.** We rely on human annotators to construct preference datasets for DPO in real-world scenarios, which is costly and time-consuming. Existing AI evaluators, however, have flawed physics knowledge and inherit biases, limiting the scalability of reinforcement learning. Fortunately, our DPO training paradigm is effective even with a small amount of human-labeled data (500 in our experiment), mitigating this limitation.

**Ethics Statement** This work has been conducted in accordance with the ICLR Code of Ethics. We have carefully considered the ethical implications of our research and can confirm several key points. The training data used in this study was sourced from open-source datasets or prepared using open-source methods. Prior to model training, we implemented a comprehensive data filtering and screening process to remove potentially harmful, biased, or problematic content. This ensures our model adheres to ethical AI principles.

**Reproducibility Statement** To ensure the reproducibility of our work, we will ensure the following points. **Code:** Our code and model will be made publicly available, including necessary scripts. **Data:** Detailed descriptions of our data construction are provided in Appendix B. **Experimental Setup:** We have stated all experimental configurations, including hyperparameters, hardware specifications in the training configuration of the main paper. **Model Architecture:** The architecture details are described in the method part, with additional details provided in the Appendix B.

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

# Appendix

# PhysMaster: Mastering Physical Representation for Video Generation via Reinforcement Learning

## A  Overall Structure

In this appendix, we first provide more details of experiments in both training and evaluation in Sec B. Next, in Sec C, we give a more detailed discussion of ablation study for data construction, training strategy and model design. Then, we show more results to validate the generalizability of our PhysMaster on real-world scenarios involving different physical knowledge in Sec D. Afterward, we present more qualitative comparisons on the proxy task and relevant scenarios in Sec E. Broader impact and limitations are discussed in Sec F and Sec G. Finally, we clarify the use of large language models (LLMs) in Sec H. We also set up **a local website** to showcase the dynamic effects of our physics-aware video generation in supplementary material.

## B  Experimental Details

**Image-to-video diffusion transformer model.**  Our PhysMaster is based on the image-to-video (I2V) task where the model is expected to predict subsequent frames from the input image.

We utilize a transformer-based diffusion (DiT) (Peebles & Xie, 2023) model as the text-to-video (T2V) foundation model and finetune the DiT model to the setting of I2V. Specifically, the DiT model employs a 3D Variational Autoencoder (VAE) (Kingma, 2013) to transform videos and initial frame from pixel space to latent space, and T5 encoder $\mathcal{E}_{T5}$ (Raffel et al., 2020) for text embeddings $c_{text}$.

We use Rectified Flow (Liu et al., 2022; Esser et al., 2024) to define a probability flow ordinary differential equation (ODE), which transfers the clean data $z_0$ to noised data $z_t$ at timestep $t$ along a straight path, formulated as:

$$z_t = (1-t)z_0 + t\epsilon, \tag{4}$$

where $\epsilon \sim \mathcal{N}(0, I)$ is Gaussian noise. Taking the first frame as image input, VAE encodes it into image embeddings $c_{image}$, which are then fed into the DiT model after being concatenated with video embeddings. We remove the noise added to the first frame of the video during training, and for inference, we replace the gaussian noise of first frame with the image embedding $c_{image}$ encoded from input image, and then predict the following frames by the model. The flow-based DiT model with weights $\theta$ directly parameterizes the $v_\theta(z_t, t, c_{text}, c_{image})$ to regress velocity $(z_1 - z_0)$ with the Flow Matching objective (Lipman et al., 2022):

$$\mathcal{L}_{LCM} = \mathbb{E}_{t,z_0,\epsilon}||v_\theta(z_t, t, c_{text}, c_{image}) - (z_1 - z_0)||_2^2. \tag{5}$$

**Training pipeline.** We train PhysEncoder on human preference data via Direct Preference Optimization (DPO) (Rafailov et al., 2023) in a three-stage training pipeline detailed in Figure 7. We first conduct supervised fine-tuning (SFT) of both base model and PhysEncoder with a global batch size of 192 and a learning rate of 1e-5. Then we adopt two-stage DPO with pairwise supervision to enhance PhysEncoder's capacity to capture physical representations and model's physical performance, with trainable module separately set as LoRA (Hu et al., 2021) of the base model and PhysEncoder independently, both with learning rate equals to 1e-6 and global batch sizes of 96 and 72. With SFT providing the model with initial ability to predict physically plausible videos guided by the simultaneously finetuned PhysEncoder, the subsequent DPO processes further steer PhysEncoder's output distribution towards physics-aware representation, thus help improve the physical accuracy of generated videos.

**Dataset construction.** Our training and test datasets of the proxy task is simulated on Kubric (Greff et al., 2022), a simulation and rendering engine designed for scalable video generation as in Figure 8. Each video consists of 1-6 dropping objects onto a (possibly empty) pile of up to 4 objects underneath them. The camera remains stationary in each video and is oriented parallel to the ground plane. The object assets are sourced from the Google Scanned Objects (GSO) dataset (Downs et al., 2022) of various true-to-scale 3D models, which are created from real-world scans across diverse

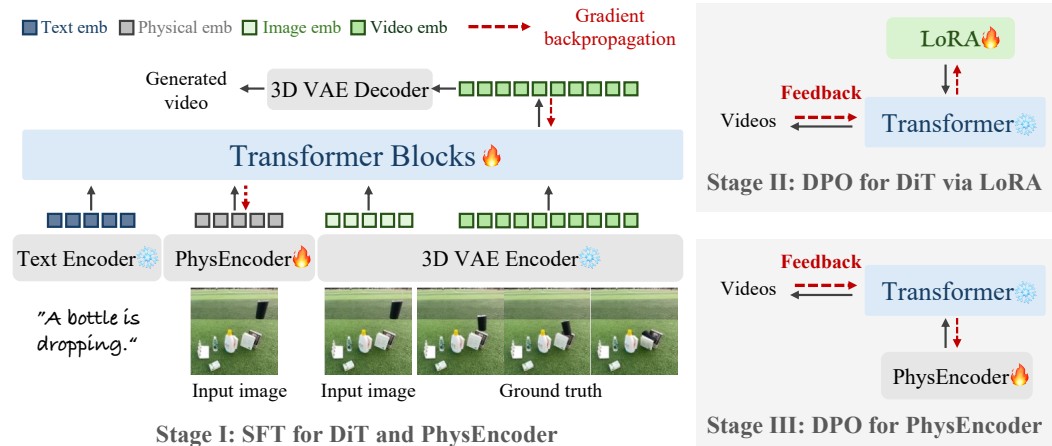

Figure 7: **Training pipeline of PhysMaster.** Given an input image, the DiT model predicts subsequent frames conditioned on physical, visual, and text embeddings. In Stage I, by concatenating physical embeddings extracted by PhysEncoder with visual embeddings encoded by VAE, we inject physical representation as extra condition to the I2V base model through SFT on both PhysEncoder and DiT model; In Stage II, we apply LoRA (Hu et al., 2021) to finetune the DiT model on preference dataset with DPO; In Stage III, we only optimize PhysEncoder 's physical representation via feedback from generated video pairs of the model in a DPO paradigm (Rafailov et al., 2023).

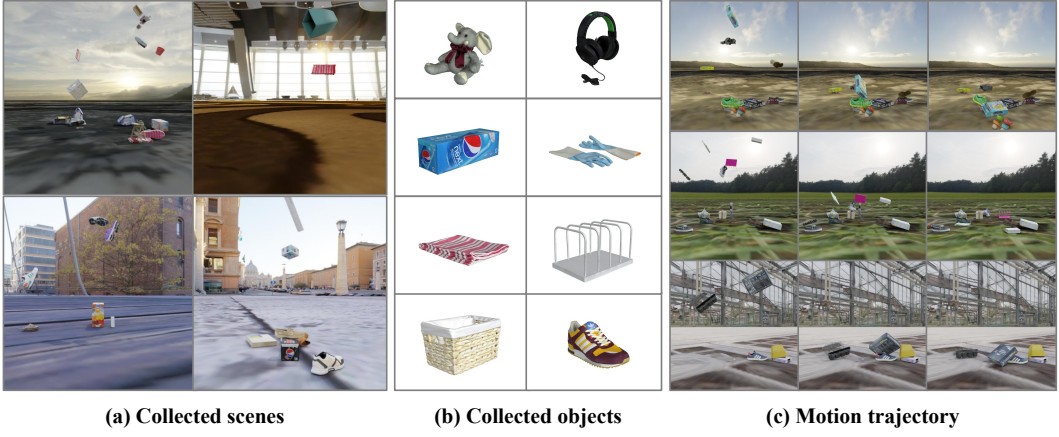

(a) Collected scenes          (b) Collected objects          (c) Motion trajectory

Figure 8: **Illustration of the dataset construction process.** We use the Kubric (Greff et al., 2022) simulation and rendering engine for creating our simulated videos. Each scene consists of objects from the Google Scanned Objects (GSO) dataset (Downs et al., 2022) and uses environmental lighting from HDRI maps provided by Kubric, the motion trajectory is simulated on PyBullet (Coumans, E. et al., 2010) and rendered by Blender (Community, B. O., 2018).

categories (e.g., shoes, vegetables, toys, .etc). Our training data for SFT in stage I comprise 7k samples and the test set includes 500 videos for preference data construction, both with resolution of 512x512. Hence, the preference datasets used in Stage II and III both consist of 500 pairs of videos with positive and negative labels. The input image for the model is sampled from the first frame of each video, and each video segment consists of 32 frames corresponding to a duration of 2 seconds at 16 frames per second (fps).

For generalizability demonstration, we utilize WISA-80K (Wang et al., 2025), a large-scale dataset collected based on qualitative physical categories and consisting of 80,000 videos, encompassing 17 types of physical events (e.g., Collision, Melting, and Reflection) across three major branches of

Table 6: **Quantitative results for models in Stage II (DPO for DiT model) using different metrics as criterion for preference data construction** on proxy task, evaluated on the test set split into "seen" and "unseen". IoU and L2 are more effective than CD with favorable model performance, which stems from their explicit correspondence to physical attributes: IoU for shape consistency, and L2 for motion trajectories.

| Preference Criterion | Seen | | | Unseen | | | Average | | |
|---|---|---|---|---|---|---|---|---|---|
| | L2($\downarrow$) | CD($\downarrow$) | IoU($\uparrow$) | L2($\downarrow$) | CD($\downarrow$) | IoU($\uparrow$) | L2($\downarrow$) | CD($\downarrow$) | IoU($\uparrow$) |
| CD | 0.0515 | 0.131 | 0.143 | 0.0481 | 0.123 | 0.142 | 0.0498 | 0.127 | 0.142 |
| L2 | **0.0513** | **0.124** | **0.145** | 0.0473 | 0.118 | **0.149** | 0.0493 | 0.121 | **0.147** |
| **IoU (Ours)** | 0.0520 | 0.125 | 0.133 | **0.0454** | **0.116** | 0.136 | **0.0487** | **0.120** | 0.134 |

physics: Dynamics, Thermodynamics, and Optics. The preference datasets used in Stage II and III both consist of 500 pairs of videos with human-annotated positive and negative labels.

**User study.** A user study is conducted to compare the physical plausibility of video pairs, separately generated by the model assisted by physical representation from PhysEncoder of Stage I and III. Specifically, 30 annotators are involved, and for the two types of real-world scenarios concerning rigid-body movements and fluid motion, 20 videos are randomly picked from the benchmark of VIDEOPHY (Bansal et al., 2024), thus a total of 40 video pairs are provided for evaluation.

## C   ABLATION STUDY

We conduct a detailed analysis of core designs of our pipeline from the model and data perspectives on the proxy task.

### C.1   DATA CONSTRUCTION

**Preference criterion.** In Stage II of DPO for the DiT model, we construct a preference dataset consisting of 500 video pairs generated by the Stage I model. Each pair contains a positive and a negative sample, determined by their similarity to the ground-truth videos. The evaluation metrics—L2, Chamfer Distance (CD), and Intersection over Union (IoU)—can all serve as similarity indicators between the generated video and the ground truth, and thus as the basis for preference assignment. To investigate the impact of these choices, we ablate the different metrics used as the decisive preference criteria for constructing these training data pairs for DPO, and evaluate the model's performance. Note that this ablation is specifically conducted in Stage II, applying DPO on the DiT model via LoRA. The results, presented in Table 6, indicate that constructing preference data using either IoU or L2 metrics leads to favorable performance. While IoU measures the spatial overlap between objects in the generated video and the ground truth, L2 represents the Euclidean distance of the centroid of object segmentation map. Compared to CD, these metrics serve as more effective standards for preference assignment, since they explicitly correspond to key physical attributes: IoU is sensitive to the changes in object properties such as shape during motion, and L2 captures the motion trajectory of free falls, making them more effective for improving the physical plausibility of generated videos.

**Data size.** We further conduct an ablation study on the size of preference dataset to investigate the data scale required for achieving optimal performance of physics-aware alignment on the benchmark. We create random subsets of 500, 1500, and 2500 samples from the full test dataset and finetune our model for the same number of steps in Stage II on each subset. Notably, as shown in Table 7, we observe that only 500 samples are needed to achieve optimal results, and they even perform better on average than larger data sizes. While a larger data scale (1500 or 2500 samples) slightly improves performance on the test split featuring objects and backgrounds "seen" during training, it significantly degrades performance on the "unseen" setting, which involves novel objects and backgrounds. This suggests that using more data pairs might lead the model to overfit to the specific domain of the training data, consequently sacrificing generalization capability to unseen scenarios.

Table 7: **Quantitative results for models in Stage II (DPO for DiT model) trained on different data sizes** on proxy task. It indicates that 500 samples suffice for optimal overall performance and even outperform larger datasets on average, and increasing data scale improves results on "seen" data but significantly degrades performance on "unseen" scenarios due to overfitting and reduced generalization.

| Data Size | Seen | | | Unseen | | | Average | | |
|---|---|---|---|---|---|---|---|---|---|
| | L2(↓) | CD(↓) | IoU(↑) | L2(↓) | CD(↓) | IoU(↑) | L2(↓) | CD(↓) | IoU(↑) |
| **500(Ours)** | 0.0520 | 0.125 | 0.133 | **0.0454** | **0.116** | **0.136** | **0.0487** | **0.120** | 0.134 |
| 1500 | **0.0508** | **0.124** | 0.136 | 0.0497 | 0.128 | 0.132 | 0.0505 | 0.126 | **0.137** |
| 2500 | 0.0514 | 0.126 | **0.142** | 0.0496 | 0.126 | 0.133 | 0.0503 | 0.126 | 0.134 |

Table 8: **Quantitative results for models in Stage II (DPO for DiT model) finetuned via different trainable modules** on proxy task, which shows that full-finetuning degrades performance compared to LoRA-finetuning, for LoRA more effectively preserving the model's intrinsic capabilities acquired during SFT in Stage I.

| Trainable Module | Seen | | | Unseen | | | Average | | |
|---|---|---|---|---|---|---|---|---|---|
| | L2(↓) | CD(↓) | IoU(↑) | L2(↓) | CD(↓) | IoU(↑) | L2(↓) | CD(↓) | IoU(↑) |
| **LoRA (Ours)** | **0.0520** | **0.125** | 0.133 | **0.0454** | **0.116** | **0.136** | **0.0487** | **0.120** | 0.134 |
| Full model | 0.0555 | 0.138 | **0.134** | 0.0501 | 0.129 | 0.134 | 0.0528 | 0.133 | **0.134** |

## C.2 TRAINING STRATEGY

**Trainable modules.** In Stage II, we perform DPO on the DiT model. We compare two finetuning strategies: full model finetuning and LoRA finetuning in Table 8, which indicates that full finetuning degrades the model's performance compared to LoRA. This finding demonstrates that applying LoRA to finetune the linear layers of the transformer model largely preserves the intrinsic performance of the model after SFT in Stage I. Consequently, this preservation leads to better physical accuracy in the generated videos, as indicated by the superior performance.

## C.3 MODEL DESIGN

**Architecture of PhysEncoder.** Our PhysEncoder comprises a DINOv2 encoder and a physical head, following the architecture of the Depth Anything model (Yang et al., 2024a). Such an architecture is denoted as "Depth-based PhysEncoder". In Stage I, the training of "Depth-based PhysEncoder" starts from a frozen DINOv2 encoder with pretrained weights from Depth Anything (Yang et al., 2024a), coupled with a trainable physical head initialized with random weights. This stage can therefore be viewed as adapting the Depth Anything architecture for the injection of physical conditions. To analyze PhysEncoder with different architectures in Stage I, we conduct SFT on both the PhysEncoder and the DiT model. Specifically, we compare our proposed architecture, "Depth-based PhysEncoder ", with the other two variants, "DINOv2-based PhysEncoder" consisting of a DINOv2 encoder pretrained from the original model and a physical head, "CLIP-based PhysEncoder" which replaces the DINOv2 encoder with a pretrained CLIP encoder. Table 9 indicates that our "Depth-based PhysEncoder" significantly outperforms "CLIP-based PhysEncoder" and holds a slight edge over the "DINOv2-based PhysEncoder". This superior performance validates the rationale behind our architectural design, demonstrating its potential for effectively capturing the essential physical conditions for our task.

## D GENERALIZATION ON REAL-WORLD SCENARIOS

Our PhysMaster demonstrates its physics-awareness by enhancing the physical realism of generated videos on synthetic data of proxy task. This suggests its potential to generalize to real-world scenarios and a broader range of physical laws. To substantiate this claim of generalization, we extend beyond "free-fall" in simulation scene by incorporating real-world scenarios. We then fine-

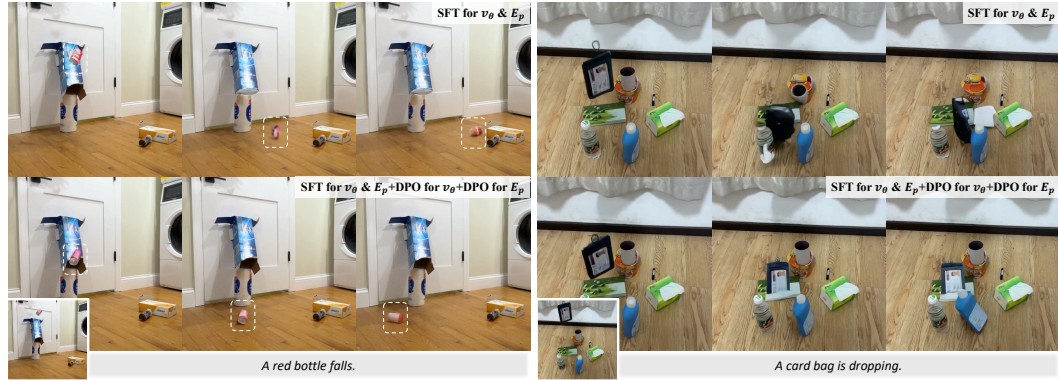

Figure 9: **Qualitative comparisons for models in each training stage on the real-world test set** of object dropping and collision. The model exhibits a preliminary capability for predicting object motion trends after SFT. Two-stage DPO further improves model performance in preserving objects' rigidity and complying with physical laws (e.g., gravitational acceleration and collision). $v_\theta$ is DiT model and $E_p$ is PhysEncoder.

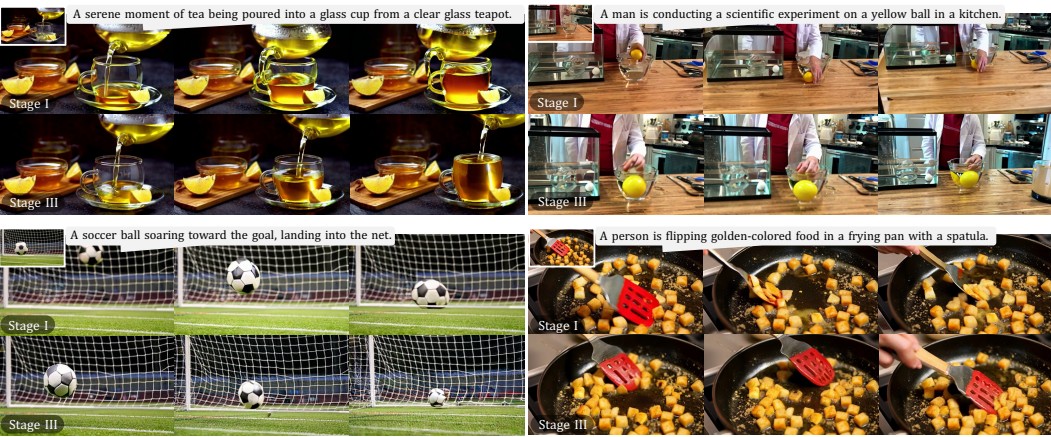

Figure 10: **Qualitative ablation for models in different stages** on broader scenarios. DPO following Stage I improves the physical coherence of model in Stage III (e.g., fluid mechanics and gravitation).

tune our model using the three-stage training pipeline on a combined dataset on both simulated and real-world data governed by different physical principles, and assess the physics-awareness of the resulting video generation model in each stage. It allows us to demonstrate the generalizability of our approach in two key aspects: (1) Physical Attributes: handling different object materials and physical laws; and (2) Data Domain: adapting to both synthetic and real-world data.

Specifically, we utilize WISA-80K (Wang et al., 2025) dataset and split it into two sets for SFT and DPO. When combining the simulated and real-world datasets for training, we introduce an extra label to prompts for distinguishing data domains: "Virtuality" for simulation data and "Reality" for real-world data. To evaluate the training results, we conduct comparisons in two aspects.

**Proxy task.** We evaluate the performance of proxy task on both real-world and simulated test datasets from PisaBench (Li et al., 2025). During inference, we also incorporate the corresponding label of "Reality" or "Virtuality" with prompts as in training. In Table 10, we quantitatively compare the performance of five model variants: row 1 is the I2V base model; row 2 - 3 refer to the models in Stage I and III of our training pipeline and row 5 - 6 are the models of comparative pipeline. "Seen" and "Unseen" represent two test splits of simulation scenario and "Real" is the real-world test dataset. The comparison indicates that in our training pipeline, with SFT endowing the model with preliminary ability to predict objects' motion "free-fall", the optimization of PhysEncoder in

Table 9: **Quantitative results for models in Stage I with PhysEncoder of different architectures** on proxy task.

| Architecture of PhysEncoder | Seen | | | Unseen | | | Average | | |
|---|---|---|---|---|---|---|---|---|---|
| | L2(↓) | CD(↓) | IoU(↑) | L2(↓) | CD(↓) | IoU(↑) | L2(↓) | CD(↓) | IoU(↑) |
| **Depth-based (Ours)** | 0.0568 | 0.141 | 0.137 | **0.0498** | **0.128** | **0.134** | **0.0533** | **0.134** | 0.135 |
| CLIP-based | 0.0580 | 0.149 | 0.129 | 0.0499 | 0.133 | 0.127 | 0.0540 | 0.141 | 0.128 |
| DINOv2-based | **0.0551** | **0.137** | **0.149** | 0.0515 | 0.132 | 0.124 | 0.0533 | 0.135 | **0.137** |

Table 10: **Quantitative results for models from different training stages and pipelines** on proxy task, evaluated on the test set split into "Seen", "Unseen" and "Real". $v_\theta$ is DiT model and $E_p$ is PhysEncoder. Our training pipeline is in the first block and the comparative pipeline is in the second.

| Training Stages | Seen | | | Unseen | | | Real | | |
|---|---|---|---|---|---|---|---|---|---|
| | L2(↓) | CD(↓) | IoU(↑) | L2(↓) | CD(↓) | IoU(↑) | L2(↓) | CD(↓) | IoU(↑) |
| Base | 0.1066 | 0.323 | 0.119 | 0.1065 | 0.339 | 0.111 | 0.1600 | 0.459 | 0.104 |
| SFT for $v_\theta$ & $E_p$ (Stage I) | 0.0543 | 0.136 | **0.158** | 0.0474 | **0.115** | **0.150** | 0.0762 | 0.179 | 0.158 |
| SFT for $v_\theta$ & $E_p$ + DPO for $v_\theta$ + DPO for $E_p$ (Stage III) | **0.0461** | **0.114** | 0.153 | 0.0466 | 0.118 | 0.145 | **0.0748** | **0.176** | **0.163** |
| SFT for $v_\theta$ | 0.0532 | 0.134 | 0.137 | 0.0512 | 0.133 | 0.135 | 0.0765 | 0.187 | 0.163 |
| SFT for $v_\theta$ + DPO for $v_\theta$ | 0.0560 | 0.143 | 0.144 | **0.0446** | 0.115 | 0.128 | 0.0755 | 0.183 | 0.163 |

last stage improves its capability in guiding model towards higher level of physics-awareness. The comparative pipeline is not equipped with PhysEncoder, thus SFT and the following DPO are both implemented on the DiT model as in the second block. It is proved that DPO of PhysEncoder unlocks the potential of approach to extract physical information and guide the model to generate videos with better physical performance by comparing row 3 and row 5. It is worth noting that, through joint training on combined data, we also achieve a significant performance enhancement on the out-of-domain real-world test data. The performance on the real-world domain does not suffer degradation, even though no real-world data for the dropping and collision task is included in the training set. This success is thanks to the domain transfer ability provided by our prompt labels and the strong generalization capability of the PhysEncoder itself. We also visualize the qualitative results in Figure 9, which consistently confirms the effectiveness of our training mechanism.

**Broader scenarios.** Figure 10 provides a qualitative comparison of the models in Stage I and Stage III of our training pipeline on real-world scenarios. The video generated by the latter model exhibits significantly more plausible physical behavior. For instance, as the liquid is poured, the liquid level in the glass bottle gradually rises, and the transparent water demonstrates realistic refraction of the ball along with believable interaction with the human hand. These results consistently demonstrate the generalizability and effectiveness of PhysMaster in injecting physical information and enhancing physical plausibility of generation across different physical phenomena and data domains. Therefore, PhysMaster provides a generalizable solution for unleashing the capabilities of physical comprehension across diverse physical phenomena. This highlights its ability to act as a foundational solution for physics-aware video generation and energize more sophisticated applications.

# E    MORE COMPARISONS

We compare qualitatively with two types of video generation models, general models including CogVideoX-5B (Yang et al., 2024c), Wan2.1-T2V-1.3B (Agarwal et al., 2025), and specialized physics-focused models represented by WISA (Wang et al., 2025). Figure 11 includes more qualitative comparison with existing T2V models, demonstrating our superior ability in challenging cases of both rigid-body and fluid motion.

# F    BROADER IMPACT

Our proposed method, which aims to achieve physics-aware video generation, holds significant potential for both positive societal impacts and necessitates careful consideration of potential negative implications. On the positive side, generating physically plausible video content can substantially enhance the realism and quality of visual media across various domains. This directly benefits con-

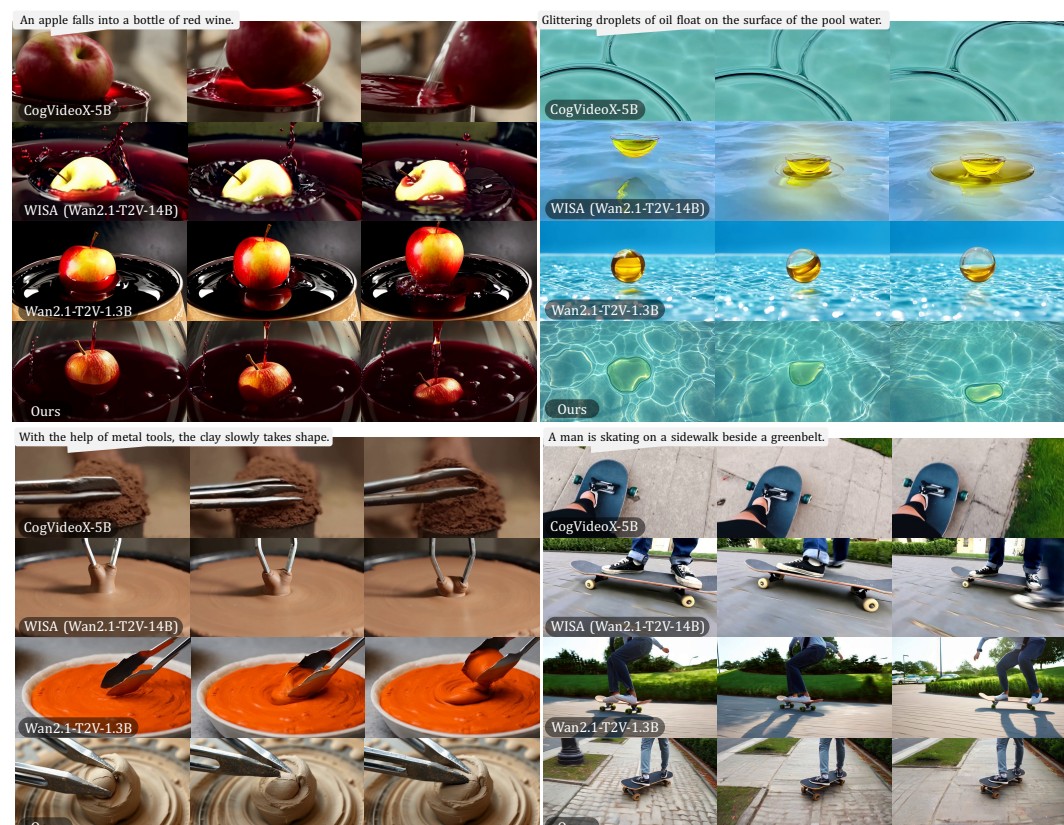

Figure 11: **Qualitative comparison with existing T2V models on broader scenarios** including objects of various materials and in different environments, validates the generalizability of our method.

tent creation for films, television, and online media by facilitating the production of more convincing visuals, thereby reducing production costs and increasing efficiency. Furthermore, the ability to generate videos adhering to physical principles is fundamentally important for predicting the motion of objects and the evolution of events in the real world. This capability is a crucial building block for realizing advanced "World Models", By serving as powerful world simulators, video generation models have broad applications in various fields, including:(1) Robotics: for training robots with generated videos that accurately reflect real-world physics, enabling safer and more efficient learning of manipulation and navigation tasks; (2) Scientific Research: for visualizing complex physical phenomena, testing hypotheses, and gaining insights into system behavior. (3) Engineering and Design: for simulating the performance of physical systems and prototypes under various conditions before physical construction. In summary, the development of more accurate and versatile World Models, enabled by physics-aware video generation, has profound implications for society.

However, it is imperative to acknowledge the potential negative impacts. As with any powerful generative technology, the ability to create highly realistic yet synthetic videos raises concerns regarding the potential misuse for generating deceptive or misleading content (e.g., deepfakes), which could erode trust and spread misinformation. Additionally, the increasing sophistication of generative models necessitates discussions around intellectual property rights and the ethical implications of creating content that may be indistinguishable from real-world recordings. Future research and deployment of such models must be accompanied by robust ethical guidelines, transparency mechanisms, and efforts to mitigate potential harms.

## G  LIMITATIONS & DISCUSSION

While our proposed method demonstrates significant potential as a generic solution for physics-aware video generation, certain limitations exist. In this study, we rely on human annotators to

construct preference datasets for DPO across a wide range of scenarios. This process, while necessary, is both expensive and time-consuming, requiring manual effort to curate diverse, high-quality data that can be used for training. However, existing scalable AI evaluators, which are designed to automate the evaluation process, often exhibit flawed physical knowledge and inherit inherent biases from their training data. These limitations restrict the overall training scale and effectiveness of reinforcement learning, as the evaluators' inaccuracies introduce noise and reduce the quality of feedback provided to the model. Fortunately, our DPO training paradigm has shown significant effectiveness even with a small amount of human-labeled data (500 examples in our experiment), alleviating the impact of this limitation on our results. To address these challenges, we plan to explore the development of more robust, physics-aware evaluators that can more accurately assess the performance of models in a wide variety of physical contexts. Such evaluators would be designed to integrate better physical reasoning and reduce biases, thereby enabling more efficient and scalable training processes. We envision that this will not only enhance the overall performance of DPO but also allow for the application of our method across a broader set of real-world scenarios, thus expanding the potential of reinforcement learning in video generation. Future research will focus on bridging this gap and advancing the state of the art in physics-aware evaluation and scalable model training.

## H    THE USE OF LARGE LANGUAGE MODELS (LLMS)

In this work, we utilized large language models (LLMs) exclusively for the purpose of grammar checking and text polishing. Specifically, LLMs were employed to assist in enhancing the clarity, coherence, and readability of the text, by identifying and correcting grammatical errors, improving sentence structure, and refining language usage. These models were not involved in any aspect of the research ideation, data analysis, experimental design, or any other stages of the research process. The content, ideas, and conclusions presented in this work are solely the result of the authors' intellectual contributions.

