# OpenReview forum: "PhysMaster: Mastering Physical Representation for Video Generation via Reinforcement Learning"
_ICLR.cc/2026/Conference — ICLR 2026 Conference Withdrawn Submission_

### Official Review · Reviewer_CGjE · 2025-10-25

**Soundness:** 3
**Presentation:** 2
**Contribution:** 3
**Rating:** 4
**Confidence:** 3

**Summary:**

This paper proposes PhysMaster, a novel framework to improve the physical plausibility of generated videos. The core idea is to introduce a PhysEncoder that extracts a "physical representation" from the initial image to guide an image-to-video (I2V) diffusion model. Since this physical representation is not explicitly defined, it cannot be directly supervised. The authors overcome this by employing a reinforcement learning approach, specifically Direct Preference Optimization (DPO), to train the PhysEncoder. The physical plausibility of the generated videos serves as feedback, optimizing the encoder to produce representations that lead to more physically realistic outcomes. The method is evaluated on a "free-fall" proxy task and then generalized to broader, more diverse physical scenarios, demonstrating improved performance over existing methods.

**Strengths:**

1. The paper's primary novelty lies in framing the learning of physical properties as a representation learning problem solved via reinforcement learning. Instead of directly finetuning the entire video model with RL or relying on explicit physics simulators, the proposed method focuses on optimizing a dedicated, plug-in PhysEncoder. This is a creative and elegant approach to inject physical knowledge into generative models.
2. The proposed three-stage training pipeline (SFT, DPO on DiT, DPO on PhysEncoder) is well-structured and methodologically sound. The experimental evaluation is comprehensive, including a controlled proxy task with clear metrics (CD, IoU) and generalization experiments on a diverse real-world dataset (WISA-80K). The ablation studies are thorough, effectively justifying key design choices like the training strategy and model components.

**Weaknesses:**

1. The learned "physical representation" remains a black box. While the PCA visualization in Figure 4 provides some insight, it is still high-level. It is difficult to ascertain whether the model is learning fundamental physical principles (like mass or friction) or simply learning complex correlations that work well for the training distribution. This makes the claim of "mastering physical representation" strong.
2. The ablation study in Table 1 (row 3 vs. row 2) shows that initial Supervised Fine-Tuning (SFT) with the PhysEncoder actually degrades performance compared to SFT on the base model alone. The paper states that DPO later "unlocks" the potential, but this initial negative result suggests the SFT stage might be suboptimal or that the integration of the encoder is non-trivial.
3. For the broader scenarios, the DPO training relies on a preference dataset of only 500 human-annotated pairs. While the authors note this as a success in data efficiency, it raises questions about the robustness and generalizability of the learned policy across the vast space of real-world physical interactions. The model might overfit to the specific types of physical phenomena present in this small dataset.

**Questions:**

1. Could you elaborate on the performance drop in Stage I when introducing the PhysEncoder (Table 1, row 3)? Why does a simple SFT objective fail to effectively utilize the PhysEncoder, and what is the intuition behind why the DPO objective is uniquely capable of correcting this and "unlocking" its potential?
2. The PhysEncoder is initialized from a DINOv2 model pretrained for depth estimation (Depth Anything). This provides a very strong geometric prior. How much of the performance gain is due to this specific initialization versus the DPO training? An ablation study with a PhysEncoder initialized from scratch or with a more general-purpose pretrained model (e.g., standard ImageNet pretraining) would help isolate the contribution of the proposed representation learning framework.
3. For the generalization experiments, the appendix mentions training on a "combined dataset" of simulated and real-world data. Could you clarify if the model is trained from scratch on this combined set, or is it finetuned from the model trained on the proxy task? How well does the physical representation learned purely on the "free-fall" proxy task transfer to the broader, more complex scenarios without this joint training?

---

### Official Review · Reviewer_sFF9 · 2025-10-29

**Soundness:** 2
**Presentation:** 2
**Contribution:** 2
**Rating:** 2
**Confidence:** 5

**Summary:**

The paper proposed to enhance video generation model (I2V) 's obedience to physics rules by introducing an aditional module (named as PhysMaster) to learn physics information implicitly from I and then inject this condition by concatenaing the physical feature after the image latent from VAE. This module is based on DINO v2 and trained by DPO RL with preference data annotated by human. Training data used is from WISA-80K. For experiment, authors compared the RL-tuned I2V model (without any detail) to other general model as well as physics-enhanced model such as WISA/PhyT2V.

**Strengths:**

The paper is clearly written, and according to the results from Table 3, it achieved better performance than existing works w.r.t. physics metrics.

**Weaknesses:**

1. The proposed PhysMaster module (initialized with DINOv2) is designed to be responsible for physics information learning. However, it only took image as input, leaving text prompt untouched. With these settings, it's highly doubtful that the information encoded by PhysMaster module is really about physics. It's simple: how can we tell what physical rules would be involved with image given only?
2. The proposed method is not training-free / plug-and-play for video generation model, this means we have to train the PhysMaster module as well as the I2V model (with lora in the paper). And since the whole model is finetuned with physics-centric data, it should definitely perform better on physics oriented metrics. This will make the contribution claims untenable.

**Questions:**

Only one addtional question: why RL here is necessary, comparing direct supervized finetuning on WISA-80K?

---

### Official Review · Reviewer_Df2J · 2025-10-31

**Soundness:** 3
**Presentation:** 3
**Contribution:** 3
**Rating:** 6
**Confidence:** 4

**Summary:**

This paper proposes PhysEncoder, a module designed to encode physical information as an additional conditioning signal, thereby injecting physical knowledge into the video generation process. Building upon this, the authors further apply the DPO algorithm to enhance the consistency between I2V-generated videos and real-world physical laws. Experimental results show that the proposed method improves video realism under both free-fall and more general scenarios, aligning generated motion with real physical conditions.

**Strengths:**

1. The paper is well-written and easy to follow.

2. The experimental section is comprehensive, and the results are convincing.

3. The visualizations in the supplementary material effectively demonstrate the qualitative improvements.

**Weaknesses:**

The actual contribution of PhysEncoder is not well demonstrated.

1. Quantitative evidence:
In Table 1, the improvement from SFT for vθ & Ep + DPO for Ep (row 6) over SFT for vθ & Ep (Stage I) is relatively limited.
The authors explain this by stating that DiT needs to align its physical perception before PhysEncoder can take effect, and they further provide Stage 3 (PhysEncoder-only optimization) as supporting evidence.
However, the number of optimization steps for each stage is not provided.
It remains unclear whether continued optimization of vθ alone could achieve similar gains, which would weaken the claimed contribution of PhysEncoder.

2. Architectural design:
It is not well justified why the structure of PhysEncoder itself enhances physical reasoning.
Could the same goal be achieved by using the latent code of the first frame with an additional projection head to model physical information more efficiently?
Further analysis or ablation is needed.

3. Feature analysis:
The role of PhysEncoder’s output tokens in visual generation is unclear.
The PCA analysis of physical features is informative, but it would be interesting to see whether similar correlations hold across the entire video sequence.
Specifically, can the physical features distinguish between free-fall and ground-contact states as the object moves?
This deserves further visualization and analysis.

**Questions:**

1. In the general-scene setting, how are the positive and negative samples constructed during the DPO stage?
What does the data distribution look like, and what are typical examples?
The authors should clarify these aspects to make the work more reproducible and easier for the community to follow.

**Details Of Ethics Concerns:**

Nan

---

### Official Review · Reviewer_aYxD · 2025-10-31

**Soundness:** 2
**Presentation:** 3
**Contribution:** 2
**Rating:** 4
**Confidence:** 4

**Summary:**

Summary:
This paper addresses the limitation of existing video generation models—their inability to adhere to physical laws despite producing visually realistic content—by proposingPhysMaster, a framework that enhances physics-awareness through learned physical representations and reinforcement learning (RL).

Contributions:
（1）Novel Physical Representation Framework: PhysMaster introduces PhysEncoder to extract implicit physical knowledge from input images, bridging the gap between visual input and physical guidance—addressing the key challenge of translating visual cues to physics-aware generation.
（2）RL-Driven Optimization for Physical Representation: Unlike prior RL methods that fine-tune the entire video model, PhysMaster uses DPO to specifically optimize PhysEncoder. This avoids overfitting to specific physical scenarios and enhances generalizability across diverse phenomena.
（3）Efficient and Generalizable Pipeline: The three-stage pipeline (SFT + two DPO stages) balances effectiveness and efficiency. It works with small human-labeled datasets (500 preference pairs suffice) and generalizes from simple "free-fall" to 17 types of real-world physical events.
（4）SOTA Performance with Practical Value: PhysMaster outperforms existing models in both physical accuracy and inference speed (generating a 5-second video in 26 seconds), making it a plug-and-play solution for physics-aware video generation and potential applications like robotics training or scientific visualization.

**Strengths:**

Strengths:
（1）PhysMaster delivers originality through targeted innovations that address prior limitations: It framesphysical representation as a plug-in module (PhysEncoder) instead of embedding physics into the generation model, decoupling physical knowledge extraction from visual generation—solving the poor generalization of simulation-based methods and overfitting of end-to-end fine-tuning.
（2）The paper maintains high methodological and experimental quality: The three-stage pipeline logically addresses sequential challenges, using well-justified components.
（3）The paper is well-organized and accessible.
（4）PhysMaster has immediate and long-term value: Its plug-in design and efficiency make it practical for content creation and prototyping.

**Weaknesses:**

Weaknesses:
（1）Over-Reliance on Human Annotation for Real-World Preference Data Limits Scalability. A core limitation of the work is its dependence on human annotators to construct preference datasets for DPO in real-world scenarios (e.g., WISA-80K). The paper acknowledges this is "costly and time-consuming", but it fails to address how this bottleneck restricts the framework’s scalability to more diverse physical phenomena (e.g., quantum mechanics, electromagnetism) or larger datasets. Worse, existing AI evaluators (e.g., VLMs) are dismissed as "flawed in physics knowledge" without any attempt to validate or improve them—a missed opportunity to reduce human effort.
（2）Lack of Analysis on PhysEncoder’s Physical Knowledge Granularity. The paper claims PhysEncoder extracts "physical priors like relative positions and potential interactions", but it provides no evidence ofwhat specific physical knowledgethe encoder actually learns.
（3）Experiments on Complex Physical Interactions Are Insufficient. The paper’s "broader scenarios" evaluation focuses on 17 physical events, but it neglectsdynamic, multi-object, or non-rigid interactions—the most challenging cases for physics-aware generation.

**Questions:**

Questions:
（1）Questions About PhysEncoder’s Learned Physical Knowledge
You state PhysEncoder extracts physical priors from input images and PCA visualization shows grouping by external forces. But do you have evidence that PhysEncoder learns causal physical principles instead of just correlational visual patterns? For example, through counterfactual testing or feature attribution to prove it captures meaningful physical laws? This clarifies if PhysMaster advances physical understanding as claimed.
（2）Questions About Preference Data Scalability
You note human annotation for DPO data is costly but dismiss AI evaluators without empirical validation. Have you tested physics-aware AI evaluators against human labels on a WISA-80K subset? What was the agreement rate? If not, why not use hybrid pipelines like AI pre-screening plus human validation to reduce effort? This addresses if the scalability bottleneck is unavoidable.
（3）Questions About Complex Physical Scenario Experiments
Your broader scenarios use WISA-80K but lack testing on dynamic multi-force or non-rigid interactions. Why were these excluded given they’re key for generalizability? For synthetic complex scenarios, have you compared generated motion to ground-truth physical quantities instead of just visual metrics? This checks if generalizability extends to challenging cases.
（4）Questions About Input Noise Robustness
All experiments use high-quality inputs, but real-world inputs have noise. Have you tested how input corruption impacts PhysEncoder’s performance? If performance degrades, have you tried lightweight preprocessing like denoising? This assesses real-world utility as a plug-in solution.

---

### Note · Authors · 2025-11-14

I have read and agree with the venue's withdrawal policy on behalf of myself and my co-authors.